# Product of Experts with LLMs:
# Boosting Performance on ARC Is a Matter of Perspective

**Daniel Franzen** [* 1 2] **Jan Disselhoff** [* 1 2] **David Hartmann** [* 3 2]

## Abstract

The Abstraction and Reasoning Corpus (ARC-AGI) poses a significant challenge for large language models (LLMs), exposing limitations in their abstract reasoning abilities. In this work, we leverage task-specific data augmentations throughout the training, generation, and scoring phases, and employ a depth-first search algorithm to generate diverse, high-probability candidate solutions. Furthermore, we utilize the LLM not only as a generator but also as a scorer, using its output probabilities to select the most promising solutions. Our method achieves a score of 71.6% (286.5/400 solved tasks) on the public ARC-AGI evaluation set, demonstrating state-of-the-art performance among publicly available approaches. While concurrent closed-source work has reported higher scores, our method distinguishes itself through its transparency, reproducibility, and remarkably low inference cost, averaging only around 2ct per task on readily available hardware.[1]

## 1. Introduction

Large Language Models (LLMs) have demonstrated extraordinary capabilities across diverse tasks, from natural language processing to code generation. Even so, evaluating the extent to which these systems possess abstract reasoning abilities continues to pose a major challenge in the artificial intelligence community. The Abstraction and Reasoning Corpus (ARC-AGI), introduced by Chollet (2019) and designed to assess core knowledge and the ability to generalize in AI, exemplifies this difficulty. Although these tasks (as

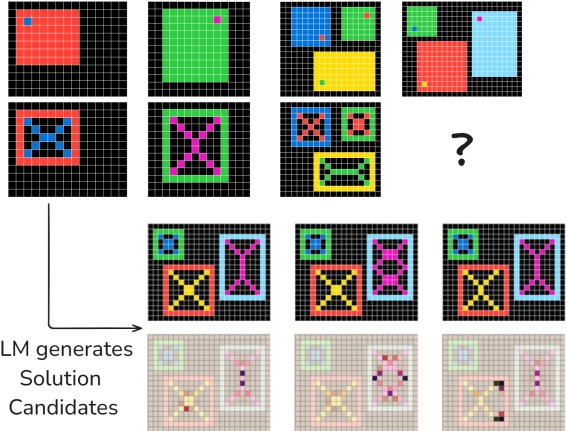

Figure 1. Example of a typical ARC-AGI task.

illustrated in Figure 1) may appear straightforward to humans, both traditional algorithmic approaches (Wind, 2020) and contemporary neural architectures (Li et al., 2024) have struggled to achieve significant success on ARC-AGI, highlighting potential limitations in current artificial reasoning methods.

Although scaling up models has undoubtedly yielded substantial performance gains on many tasks, size alone does not fully address the core limitations evident in challenges like ARC-AGI. Indeed, the rapid evolution of open-source systems – such as LLaMA-3.2-3B (Dubey et al., 2024) and Nvidia NeMo-Minitron-8B (Sreenivas et al., 2024) – demonstrates that significant capabilities can emerge even at more modest scales. This aligns with mounting evidence that many perceived shortcomings in large language models stem from implementation details or suboptimal data representations rather than from fundamental reasoning deficits (Singh & Strouse, 2024; Bostrom & Durrett, 2020; Sun et al., 2023). For instance, Allen-Zhu & Li (2025) observe that models may be aware of their mistakes without being able to correct them, while Allen-Zhu & Li (2024) highlight how subtle data modeling choices can impede fine-tuning progress. Collectively, these insights suggest that models often possess the latent capacities needed to tackle ARC-AGI;

*Equal contribution [1]Johannes Gutenberg University Mainz [2]Members of "the ARChitects" Kaggle team. [3]Lambda, Inc.. Correspondence to: Daniel Franzen <dfranzen.it@gmail.com>, Jan Disselhoff <JanDissel.it@gmail.com>, David Hartmann <davidh@lambda.ai>.

*Proceedings of the 42nd International Conference on Machine Learning*, Vancouver, Canada. PMLR 267, 2025. Copyright 2025 by the author(s).

[1]We assume a price of 36ct/hour for a Nvidia 4090 GPU

*Table 1.* Performance comparison of related work. We distinguish between solutions where the underlying model weights are open-source or proprietary.

| Model Name | Public Eval Accuracy [%] | Open Source |
|---|---|---|
| o1-preview (Kamradt, 2024) | 21 | ✗ |
| Ryan Greenblatt (Greenblatt, 2024) | 42 | ✗ |
| Jeremy Berman (Berman, 2024) | 58.5 | ✗ |
| **GPT o3** (arcprize.org, 2025) | **82.8** | ✗ |
| Avg. Human (LeGris et al., 2024) | 60.2 | ? |
| TTT (Akyürek et al., 2024) | 53.5 | ✓ |
| BARC (Li et al., 2025) | 56.75 | ✓ |
| TTT+BARC (Akyürek et al., 2024) | 62.8 | ✓ |
| **Ours** | **71.6** | ✓ |

the real challenge is creating the conditions under which these capacities can be reliably expressed.

Building on these insights, we developed an approach specifically tailored to the ARC dataset. Our method achieves SOTA performance for open source models of 71.6% (or points) on the public ARC-AGI evaluation set and surpasses average human performance of 60.2%, as measured by LeGris et al. (2024).

In particular, we employ a depth-first search (DFS) algorithm on LLM predictions to generate diverse, high-probability solutions, and re-use the same LLM also as a *product of experts* (see Section 4.1) to select the best candidate. This dual role allows us to rank candidate solutions via augmented likelihood estimates, effectively amplifying the model's latent reasoning abilities. Compared to more heavily scaled or closed-source systems, our method stands out for its transparency, reproducibility, and low inference cost of around 0.02$ per task, in stark comparison to 17$ per task for o3 (arcprize.org, 2025). This demonstrates that abstract reasoning on ARC-AGI is not exclusively the domain of massive proprietary models.

In the sections that follow, we detail our data modeling and training strategies, describe our DFS-based solution exploration, and provide comprehensive results and ablation studies.

Our final model, along with the training and inference code, is publicly available on GitHub.

## 2. Related Work

The Abstraction and Reasoning Corpus (ARC) has played a central role in advancing research on abstract reasoning in artificial intelligence, inspiring a wide range of studies focused on its dataset, competitive benchmarks, and the development of solutions driven by resource constraints.

**Visual Representation of a Task Instance:**

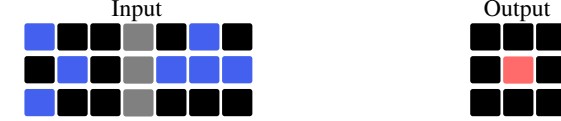

**Compact String Format of same Instance:**

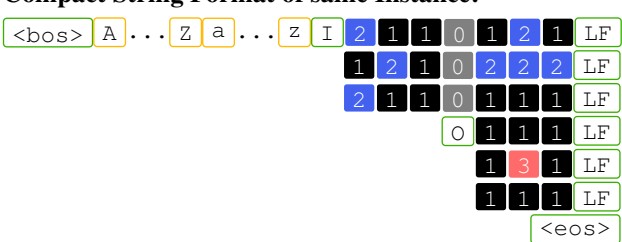

*Figure 2.* Our standard tokenization approach. Note that we use one token per cell instead of compressing the problem more. We also try to not include any unnecessary delimiters. The Pre-prompt (the alphabet in upper then lower case, i.e. "A...Za...z") is only included for the first example. Depending on the model and run there might be some small changes to the pre-prompt and input/output prefix tokens.

**The Original ARC Dataset:** The Abstraction and Reasoning Corpus (ARC-AGI) introduced by Chollet (2019) challenges the idea that language models can efficiently generalize from a small number of examples, often referred to as *few-shot prompting*. The original ARC-AGI dataset consists of 900 reasoning tasks, divided into 400 training tasks, 400 public evaluation tasks, and 100 private and thus unpublished evaluation tasks. Each task involves input and output grids of varying sizes, ranging from 1x1 to 30x30 and utilize a palette of ten distinct colors.

The objective of each individual ARC task is to discern the transformation rule from input to output from the examples and apply it to new input grids to generate the correct outputs. A task is considered successfully solved when the model produces the accurate output within a maximum of two attempts. Designed to be straightforward for humans yet challenging for machine learning systems, the tasks highlight the current limitations of AI in abstract reasoning. In a study by LeGris et al. (2024), the average human was able to correctly solve 60.2% of the evaluation tasks, while 97.8% of the tasks were solved by at least one participant using two guesses.

**Competition-driven Progresses:** Since ARC's introduction in 2019, several competitions with hundreds of participants have sought to develop solutions with strong performance on the dataset. Approaches up to 2024 frequently employed program search over domain-specific languages (DSLs), and have yielded a score of 39% using Top-3 scoring (Wind, 2020).

In 2024, ARC-AGI hosted another Kaggle competition, where for the first time large language model (LLM) ap-

proaches dominated the leaderboard. One popular method was **test-time training (TTT)**. This approach was first introduced in Sun et al. (2020), first suggested for ARC by Cole (2024) and later popularized by Akyürek et al. (2024). Test-time training leverages the few examples provided in each challenge as a small dataset. By fine-tuning on these examples before generating an answer, LLMs can achieve a substantial increase in performance. In Akyürek et al. (2024), the authors demonstrate that TTT more than doubles their performance on ARC-AGI. TTT is particularly effective in competition settings like ARC, as it allows models to extract additional training data from the limited examples available, enhancing their ability to generalize and solve new tasks.

**Notable Mentions:** Other approaches explored various strategies for utilizing LLMs. In Li et al. (2025), the authors classify two different avenues: *Induction*, where a LLM infers a function that can solve the problem which is then applied (often using python or a DSL), and *Transduction*, where the LLM directly generates the solution using a tokenized description of the problem (see Figure 2). The authors argue that these approaches solve different kinds of problems, despite using the same underlying architecture. In their experiments, induction and transduction solve roughly the same amount of problems ($38\%$ and $43\%$ respectively), which can be increased to $56.75\%$ by employing ensembles. Additionally, they use the induction network to generate a large set of novel challenges, dubbed ARC-Heavy. Some approaches make use of alternative ARC datasets, such as ConceptARC (Moskvichev et al., 2023). The most notable - and only additional dataset we use - is the well-known RE-ARC dataset. Hodel (2024) introduced this dataset, which implements generators for all 400 tasks of the public training dataset. Their code can be used to produce an arbitrary amount of training data for these tasks, but does not introduce novel challenges. All other datasets might include challenges that mimic the evaluation challenges - thereby reducing the difficulty of those challenges immensely. By only using the RE-ARC dataset, we still increase our training data immensely, but stay close to solving the ARC challenge as intended.

Data augmentation has been a common approach in previous ARC-AGI competitions (Akyürek et al., 2024; Li et al., 2025). However, our method extends beyond traditional dataset augmentation, applying transformations throughout our approach, during training (initial finetuning as well as test time training), inference and selection.

Table 1 compares recent ARC approaches, revealing that OpenAI-o3, a closed-source method, currently reports the highest score but lacks reproducible details. Further, o3 uses an immense amount of computation for each task, using 17\$ of compute for a single challenge (arcprize.org, 2025).

In contrast, TTT+BARC is fully open-source and notably the first public approach to surpass the average human performance on ARC, showcasing the benefits of transparent methodology in advancing abstract reasoning research.

## 3. Notations and Setup

To ground our approach formally, we adopt a Bayesian perspective on puzzle-solving, treating each puzzle as a partial observation from an underlying distribution of solutions.

We consider a collection of tasks (for example drawn from the ARC benchmark), where each task is denoted by $p \in \mathcal{P}$, and $\mathcal{P}$ represents the space of all possible tasks. For each task $p$, there exists an associated *solution space* $\mathcal{S}_p$.

**Problem Representation.** Throughout this paper, we use the terms *task*, *puzzle*, and *problem* interchangeably, all referring to a specification given by a small set of $k$ input-output examples and a single test input. Concretely, we write

$$p = \left( (x_i, y_i)_{i=1}^k, \, \hat{x} \right),$$

where $(x_i, y_i)$ indicates the $i$th input-output example pair and $\hat{x}$ is the test input for which we seek the correct output. Although not explicitly observed, each problem $p$ admits at least one *correct solution* $s_p^* \in \mathcal{S}_p$.

We assume the existence of a true probability distribution

$$P(s \mid p)$$

over candidate solutions $s \in \mathcal{S}_p$. If exactly one possible valid answer exists the distribution $P(\cdot \mid p)$ would be sharply peaked at $s_p^*$. While this is the case for most challenges, we will keep our theory more general, assuming multiple valid answers might exist. This can also arise from insufficient information in the given example pairs, which in the worst case prevents us from uniquely inferring the correct solution based solely on the provided data. Examples for this are sometimes found in ARC-AGI, which frequently results in an update of the dataset (Neoneye, 2024; RubenKelevra, 2024).

Hence, $P(\cdot \mid p)$ may be spread out over several plausible hypotheses. Identifying $s_p^*$ from $\mathcal{S}_p$ typically requires leveraging priors or additional constraints (e.g., knowledge of how ARC tasks are designed). Formally, one may write a posterior

$$P(s \mid p) = \frac{P(p \mid s)\, P(s)}{P(p)},$$

where $P(s)$ encodes how we believe solutions are structured *a priori*, and $P(p \mid s)$ measures how well $s$ explains the limited observed examples. The goal is to select

$$s_p^* = \underset{s \in \mathcal{S}_p}{\operatorname{argmax}} \, P(s \mid p),$$

**Search Performance Comparison**

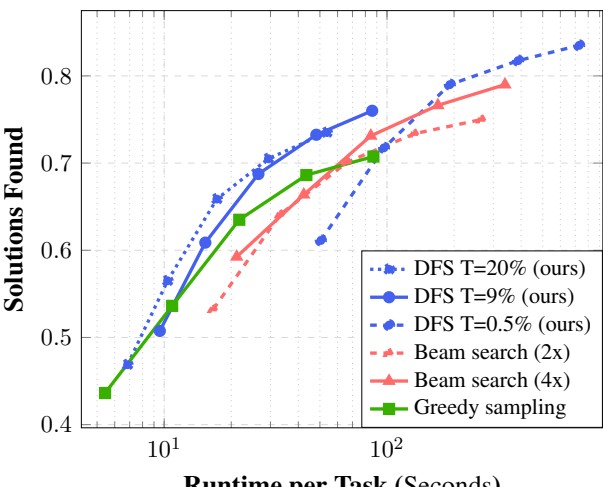

*Figure 3.* Number of solutions found by various sampling algorithms as a function of runtime. The different values for each sampling variant are calculated using 1 (identity), 2 (reflections), 4 (rotation), 8 (reflections+rotation) and 16 augmentations. Additionally, colors and the order or examples are randomly permuted in each augmented version of a task. For almost any runtime budget, we find that a DFS variant discovers the most solutions.

but in practice we do not have direct access to $P$. Instead, we train a model to approximate it, yielding $\hat{P}$ as a stand-in for the true distribution.

Finally we define a family of problem transformations ("augmentations"),

$$\Phi = \{\phi_1, \ldots, \phi_m\},$$

where each $\phi_j$ transforms both a problem $p$ and its solutions $s$ such that

$$P(s \mid p) \;=\; P\big(\phi_j(s) \,\big|\, \phi_j(p)\big) \quad \text{for all } (p, s).$$

For the ARC puzzles, such augmentations include rotations and reflections of each task, shuffling of the example order and permutation of colors. The augmentations in $\Phi$ define parts of the prior $P(s)$ by encoding invariances that are expected to hold for all valid solutions.

## 4. Methods

Our approach trains a large language model (LLM) to approximate the true solution distribution $P(\cdot \mid p)$. Given a task $p$ and some solution candidate $s$, we tokenize both and use the trained LLM to calculate probabilities for each token. By aggregating the probabilities of the solution tokens, we can define a probability function $\hat{P}(s \mid p)$ that describes the probability of sampling $s$ given $p$ as a prefix, setting the stage for subsequent sampling and search-based refinement.

**Accuracy for Different Selection Methods**

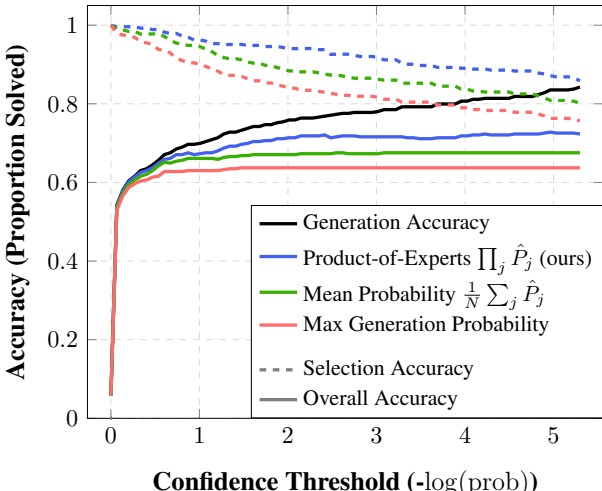

*Figure 4.* Top-2 accuracy and coverage of different selection methods as a function of the confidence threshold $T$. Solid colored lines denote the fraction of tasks solved using a specific selection algorithm. The solid black line shows the fraction of tasks where the correct solution was among the sampled candidates, and thereby provides an upper bound for the performance of the selection algorithms. The dotted lines evaluate the performance of our selection algorithms, compared to this upper bound: What percentage of correct candidates are actually selected when they are present? It shows that even when using low DFS probabilities - and therefore sampling a high number of candidates - PoE is able to select the correct solution among all candidates with high specificity.

**(1) Augmentations.** While naive multinomial sampling from $\hat{P}$ can already produce favorable candidate solutions, we enhance the model's robustness further by leveraging *augmented* training data.

These augmentations diversify the training distribution without altering correct solutions, effectively shaping the model's learned prior.

The trained LLM then provides us with a probability distribution $\hat{P}(\cdot \mid p)$ over solutions $\mathcal{S}_p$. Using multinomial sampling, this allows us to sample $s \sim \hat{P}(\cdot \mid p)$ However, sampling repeatedly from $\hat{P}$ may be expensive and does not ensure coverage of high-probability solutions. In contrast, enumerating $\mathcal{S}_p$ in full would provide us with full knowledge of $\hat{P}(s)$ but is impractical. Instead, we rely on a more systematic procedure to select promising candidates by deriving a *candidate set* of high-probability solutions via threshold-based search. Subsequently, we refine their probabilities by aggregating over multiple problem augmentations.

**(2) Candidate Generation.** To address the mentioned challenges of multinomial sampling, we propose a threshold-based search mechanism. Instead of mere random sampling,

**Correct Solution Rank Improvement**

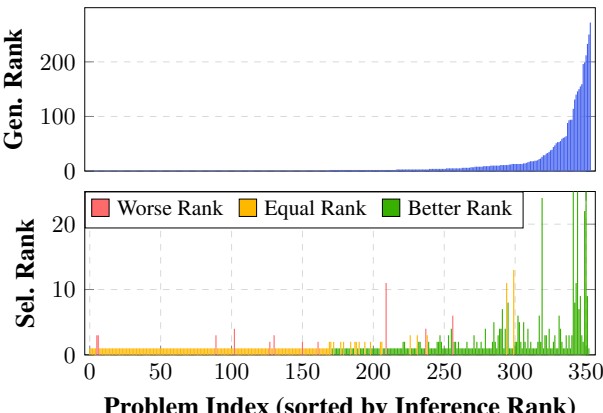

*Figure 5.* Comparing the rank of the correct solution using the generative model $\hat{P}$ and the ensemble selection $\overline{P}$ among candidates $\mathcal{C}_{p,T}$. If possible, our ensemble almost always improves the rank of the correct solution, increasing the chance of selecting it. For readability we clip the lower plot at a rank of 25.

we systematically explore the space of solutions via a **depth-first search (DFS)** algorithm.

Given a test problem $p$, we derive a set of *candidate solutions* by sampling under all valid augmentations $\phi_j(p)$. Concretely, we define

$$\mathcal{C}_{p,T} := \left\{ s \in \mathcal{S}_p \mid \exists \phi_j \in \Phi : \hat{P}\big(\phi_j(s) \mid \phi_j(p)\big) > T \right\},$$

where $T > 0$ is a threshold on the LLM's probability estimates. In practice, we run a Depth-First Search over the space of potential solutions, pruning any partial path whose accumulated probability falls below $T$. If multiple augmentations yield the same solution (up to augmentation), we merge them into a single candidate. By caching intermediate computations during inference, this DFS-based approach can rapidly pinpoint *all* likely solutions above the threshold $T$. This guarantees that solutions with sufficiently high $\hat{P}(s \mid p)$ are not overlooked and solutions with low $\hat{P}(s \mid p)$ are never considered.

**(3) Candidate Ranking via Product of Experts**  However, once we have generated the set $\mathcal{C}_{p,T}$, the highest-probability solution according to a single augmentation is not always correct.

This limitation is partially caused by the autoregressive architecture, as models can only attend to previously generated tokens when predicting the next token. This constraint means optimal decisions sometimes require information that becomes available only in later predictions. In the Sudoku experiment (Section 5.6), for instance, the model may need to solve the entire puzzle internally before predicting the first cell, potentially leading to confident but incorrect early predictions. Once an error occurs, the model cannot recover

since subsequent predictions build on the incorrect foundation. The model lacks training for stability under prediction errors, causing cascading mistakes with unexpectedly high confidence scores. This explains why the highest probability sequence from a single forward pass may not correspond to the globally optimal solution.

We can mitigate this issue, and even benefit from it, by re-augmenting each candidate $s$ under every $\phi_j \in \Phi$ and computing its likelihood using $\hat{P}$ provided by the LLM. Unlike in the previous step, this phase does not rely on generative sampling; instead, it directly evaluates the log-likelihood of $s$'s tokens for each augmented input $\phi_j(p)$. Using these re-augmented candidates, we form a single aggregate score by taking the product of probabilities across all augmentations:

$$\text{score}_{\text{agg}}(s) = \prod_{\phi_j \in \Phi} \hat{P}\big(\phi_j(s) \mid \phi_j(p)\big).$$

This product-based approach is sensitive to outliers, filtering solutions that seem unlikely from a different augmentation perspective. As a result, this approach, on average, outperforms a randomly selected augmentation, as we prove in the following section. Finally, we select the solution

$$s_p^* = \underset{s \in \mathcal{C}_{p,T}}{\operatorname{argmax}} \text{score}_{\text{agg}}(s),$$

as the final answer for problem $p$.

This two-step approach; (1) DFS-based generation with single-augmentation pruning and (2) post-hoc multi-augmentation scoring, ensures that we systematically explore high-probability solutions and then refine their rankings, accounting for LLM inconsistencies across problem representations. In practice, even if several solutions enter $\mathcal{C}_{p,T}$, their final ranks can vary greatly. By consolidating evidence from multiple perspectives, the correct solution often stands out and becomes easier to pinpoint.

### 4.1. Product of Expert Augmentations

We next analyze the performance of our ensemble method in terms of the KL divergence of the ensemble distribution $\overline{P}$ compared to the true distribution $P$. Let each valid augmentation $\phi_j$ induce approximations of the true augmented distributions $P(\phi_j(s) \mid \phi_j(p))$ by the LLM, denoted as

$$\hat{P}_j(s) := \hat{P}\big(\phi_j(s) \mid \phi_j(p)\big).$$

Since the LLM may be inconsistent across augmentations (in contrast to the true distribution $P$), our approach described in the last section combines them by the *geometric-mean ensemble*:

$$\overline{P}(s) := \frac{1}{Z} \prod_{j=1}^{m} \big[\hat{P}_j(s)\big]^{\frac{1}{m}},$$

where $Z$ is the normalization constant. A value of $Z = 1$ represents the case that the LLM is consistent across all augmentations, $P_i = P_j$ for all $(i, j)$. Intuitively, $\overline{P}$ places low probability on those $s$ for which even a few $\hat{P}_j(s)$ are low probability.

We aim to show that if each $\hat{P}_j$ is close to the *true* distribution $P$ in terms of KL divergence, then $\overline{P}$ provides - in expectation - a better estimate of $P$ than any randomly chosen $\hat{P}_j$. We formalize this idea in the following well-established theorem known from literature (Hinton, 1999; 2002).

**Theorem 4.1** (Error Bound for Log-Pooled Augmentations). *Suppose we have $m$ valid augmentations $\{\phi_1, \ldots, \phi_m\}$ in the sense of preserving solution distribution, and define*

$$\hat{P}_j(s) := \hat{P}\big(\phi_j(s) \mid \phi_j(p)\big), \text{ for each } j = 1, \ldots, m.$$

*Assume each single-augmentation predictor $\hat{P}_j$ has a bounded KL divergence from $P$, i.e.,*

$$D_j := \text{KL}\big(P \,\|\, \hat{P}_j\big) \leq \delta_j.$$

*Now define the "geometric-mean" ensemble*

$$\overline{P}(s) := \frac{1}{Z} \prod_{j=1}^{m} \big[\hat{P}_j(s)\big]^{\frac{1}{m}},$$

*where*

$$Z = \sum_{u \in \mathcal{S}_p} \prod_{j=1}^{m} \big[\hat{P}_j(s)\big]^{\frac{1}{m}},$$

*Then the KL divergence between $P$ and $\overline{P}$ is given by the average of the single-augmentation divergences and $Z$:*

$$\text{KL}\big(P \,\big\|\, \overline{P}\big) = \frac{1}{m} \sum_{j=1}^{m} \text{KL}\big(P \,\|\, \hat{P}_j\big) + \log Z$$

*With $\log Z \leq 0$, and equality iff $\hat{P}_i = \hat{P}_j$ for all $i, j$.*

See Appendix C for a proof of Theorem 4.1. The key takeaway is that $\log Z \leq 0$ becomes smaller whenever augmentations disagree, which can *improve* the ensemble in expectation relative to any random single-augmentation predictor. As a result, this approach performs especially well when different experts disagree - a state which naturally arises in our case, due to the causal autoregressive nature of the LLMs.

**Practical Implications** In practice, a product of experts approach often shines when different augmentations catch different errors. As long as the true solution does not get zero probability under any single augmentation, it remains viable. Hence, while disagreements between augmentations can prune out plausible-but-incorrect candidates, correct ones accumulate strength across viewpoints. This synergy typically yields more reliable predictions than relying on a single representation of the problem alone.

## 5. Experiments

Our approach to solving ARC-AGI combines data expansion, multi-stage fine-tuning of language models, and specialized solution evaluation. Below, we explain how these components work together to improve the model's performance while keeping computational costs manageable.

### 5.1. Data Modeling

In order to apply LLMs to ARC-AGI puzzles, we need to tokenize the data in a manner suitable for our model. This process requires careful consideration of two main challenges:

First, due to the limited context size in typical LLM architectures, an increase of inference time and decline in performance on long context tasks (Liu et al., 2024), we require a representation that minimizes the number of tokens the model needs to process. Secondly, it is widely recognized that numerous common failure modes in Large Language Models (LLMs) stem from tokenization (Singh & Strouse, 2024; Bostrom & Durrett, 2020; Sun et al., 2023). For instance, standard tokenization techniques group numbers (some but not all combinations) of one, two or three succeeding digits into dedicated "grouped-digit tokens" (Singh & Strouse, 2024). These kinds of merges would complicate the puzzles unnecessarily.

To address this, we opted to simplify the token set available to the model. In particular, we reduced the number of tokens available from over 120.000 to 64 tokens (see Table 5 in the Appendix).
This reduction offers key benefits. It significantly decreases the model size, as we can remove the majority of rows from the embedding layer. Further, token merges that typically occur during text tokenization are no longer possible. This ensures that the model can focus precisely on the data without the interference of digit separators.
As illustrated in Figure 2, we add a small number of extra tokens to the start of a task. Surprisingly, this addition slightly improves the model's performance. We believe that during fine-tuning (where the embedding layers are also trained), the model learns to use these extra tokens as a form of computational buffer, which influences every subsequent token, thereby enhancing overall performance.

### 5.2. Training the models

Choosing a suitable large language model (LLM) was essential for achieving strong performance. After evaluating

*Table 2.* Two-guess-accuracy on the ARC-AGI public evaluation set when adding parts of our method. Baseline score shows performance of our network after initial fine-tuning, generating two samples with stochastic sampling. TTT adds test-time training. 16xAug samples one solution candidate for each of 16 random augmentations of each task, choosing the two with highest sampling probability as guesses. PoE uses the product of experts to select the two best of the 16 sampled candidates, again using 16 (different) random augmentations to calculate the PoE score. Finally, DFS leverages our custom depth-first-search sampling scheme with $T = 9\%$ for candidate generation.

| Model | Baseline | + TTT | + 16xAug | + PoE | + DFS |
|---|---|---|---|---|---|
| Llama-3.2-3B | 14.9% | 40.9% | 52.9% | 59.5% | 61.4% |
| NeMo-Minitron-8B | **18.3%** | **44.5%** | **62.5%** | **67.6%** | **71.6%** |

various models, we identified **Mistral-NeMo-Minitron-8B-Base** (Sreenivas et al., 2024) as exhibiting the strongest performance in our experiments. Given the model's size, efficient fine-tuning methods were necessary for effective utilization.

Therefore, we used Low-Rank Adaptation (LoRA) (Hu et al., 2022), 4-bit quantization and gradient checkpointing, all supported by the unsloth library. We applied the LoRA adaptations to all layers of the network, including the input and output embeddings.

For each task

$$ p = \big( (x_i, y_i)_{i=1}^k, \ \hat{x} \big), $$

with solution $s_p^*$, we computed gradients only on the outputs $y_i$ for $i > 1$ and $s_p^*$. This approach ensures that the model is never tasked with predicting an input grid, and acknowledges that correctly predicting the first output grid is impossible without at least one example. To increase the amount of training data, and to better align the LLM with the data prior, we train on augmented data, adding all $D_8$ symmetries of any given task as well as color permutations and re-ordering of the examples.

**Initial fine-tuning:** The initial fine-tuning used a LoRA rank of 256 and was done on a single H100 GPU. While several ARC-like datasets exist, such as ConceptARC (Moskvichev et al., 2023) and ARC-Heavy (Li et al., 2025), we elect to only use RE-ARC (Hodel, 2024) for training. This is done to minimize "conceptual leakage", where a particular type of problem might be present in the training data, reducing the difficulty of the evaluation tasks substantially in a way that was not intended. Instead, we train only on replications of the training examples of the offical ARC-AGI training set (i.e. RE-ARC), minimizing this effect and making sure that our results are robust.

**Test-time training:** Secondary training was time-constrained and focused solely on the evaluation set, using a LoRA rank of 32 and running for 64 training steps with batch size 1. Just using test-time training increases the percentage of correctly solved tasks significantly, as can be seen in Table 2. Varying training parameters only had marginal effects.

The initial fine-tuning took 98 GPU hours on a Nvidia H100, while test-time training takes (on average) 51 seconds for a single task on a Nvidia RTX 4090 GPU. For an overview of our training parameters, see Table 4 in the Appendix.

### 5.3. Solution Inference

As introduced in Section 4, we generate potential solution candidates using DFS-based sampling to produce the set $C_{p,T}$. The goal here is to generate a small set of candidates with a high chance of containing the correct solution - and doing so quickly. Our set of augmentations $\Phi$ includes 16 functions per task - each $D_8$ symmetry is used twice but with different, randomly chosen color permutations and example re-orderings. Note that this is the same *class* of augmentations as used in training, but each color permutation and example ordering is newly randomized. Table 3 provides a comparison between DFS, Beam-search, multinomial and greedy sampling. DFS sampling is able to quickly and efficiently find a high quality set of candidates, while having low computational overhead compared to stochastic sampling for generating multiple solutions and using substantially less VRAM than beam search. In addition, it exhibits a lower false positive rate. While DFS with $T = 9\%$ finds less correct solutions than 4x Stochastic sampling (76.0% vs 77.3%), it still results in a better selection score, as it, on average, only returns about half as many false positives. Moreover, DFS accomplishes this using only a fourth of the inference time (9:32h vs 39:47h).

**Comparison to beam search:** While beam search with 4 beams can achieve the same accuracy as DFS with $T = 9\%$, it requires roughly twice the amount of VRAM (7.3GB vs 14GB), as it explores four paths simultaneously, while DFS only needs to keep a single path in memory at any time. It also takes four times as long (37:36h vs 9:32h) for the candidate generation step. The speed advantage of DFS comes mostly from early pruning of low probabilty paths. In Beam Search, the same amount of paths is explored each time, regardless of their cumulative sampling probability, while DFS stops when the cumulative sampling probability falls beyond the chosen threshold, thereby reducing unnecessary computations. Additionally, for all augmentations after the first one, we pass the most promising solution candidate

*Table 3.* Comparison of sampling and selection strategies on the 400 tasks of the ARC-AGI public evaluation set: Under "Candidate generation", we list the percentage of correct solutions sampled with different strategies using 16 augmented versions (reflections, rotations, and randomly permuted colors and examples) of each task. We also list the average number of candidates generated per task, the runtime of the sampling process on the full dataset and the maximum video memory consumption. Under "Selection", we compare the accuracy of various selections strategies, performed on the scores calculated in a subsequent scoring process on 16 additional random augmentations. Total runtime includes the test-time fine-tuning on a task's examples (see Table 4 in the appendix), as well as the candidate generation and selection process. All experiments were performed with the NeMo-Minitron-8B model on a Nvidia RTX 4090 GPU.

| Sampling method | Candidate generation | | | | Selection (2-guess accuracy) using 16 augmentations | | | | Total runtime |
| | solutions found | avg. cand. per task | runtime [hh:mm] | max. VRAM | $\max \hat{P}_j$ | $\min \hat{P}_j$ | $\sum \hat{P}_j$ | $\prod \hat{P}_j$ | [hh:mm] |
|---|---|---|---|---|---|---|---|---|---|
| Greedy | 70.8% | 6.7 | 9:39 | 7.0 GB | 63.3% | 65.8% | 66.1% | 67.6% | 18:52 |
| Stochastic (2x) | 74.5% | 11.2 | 19:53 | 7.0 GB | 64.5% | 67.6% | 66.9% | 69.9% | 34:08 |
| Stochastic (4x) | 77.3% | 17.6 | 39:47 | 7.0 GB | 63.5% | 68.8% | 67.1% | 70.8% | 58:55 |
| Beam search (2x) | 75.0% | 15.9 | 29:33 | 9.6 GB | 63.1% | 65.9% | 65.0% | 69.9% | 47:27 |
| Beam search (4x) | 79.0% | 34.7 | 37:36 | 14.0 GB | 61.9% | 67.9% | 65.0% | **71.6%** | 71:39 |
| DFS T=20% (ours) | 73.5% | 4.9 | **5:58** | 7.3 GB | 63.5% | 68.1% | 66.4% | 70.0% | **14:12** |
| DFS T=9% (ours) | 76.0% | 9.3 | 9:32 | 7.3 GB | 63.5% | 68.8% | 66.6% | **71.6%** | 20:50 |
| DFS T=0.5% (ours) | **83.5%** | 84.7 | 80:56 | 7.3 GB | 63.3% | 69.1% | 66.9% | **71.8%** | 134:43 |

found so far as an inital guess to the DFS and process it in a single forward pass before starting backtracking, which is much faster than token-by-token generation. Note that these comparisons should be interpreted with caution, as the beam search algorithm is not implemented in the unsloth library used for the other experiments, which might provide some time savings. However, even when accounting for those savings, beam search still requires far more time overall, as it returns a significant amount of false positives, which increase the runtime required in the subsequent selection process, where each candidate is evaluated under different augmentations.

As we do not know the sampling probability of the correct solution beforehand, we have to treat the probability bound $T$ as a hyper-parameter. We found that values between $T = 5\%$ to $T = 20\%$ provided a reasonable compromise between inference time and number of correct solutions, but the exact parameter depends on the model and training procedure used. Similarly, due to the way probability mass is distributed on the solution tree, DFS is faster when the model has a higher degree of certainty in its predictions.

We compare the number of candidate sets that contain the correct solution for different values of $T$ in Figure 4. This function is monotonically increasing in $T$, but so are inference costs and the size of the set $C_{p,T}$, making the selection of the correct candidate harder. Our final results are calculated using $T = 9\%$, as it uses roughly the same amount of inference time as greedy sampling.

We provide pseudo-code for our DFS sampling algorithm in Algorithm 1 in the Appendix.

### 5.4. Selection Strategies

Up to this point, our method generates candidates likely to include the correct solution. However, solving the task requires identifying it among the candidates, using at most two guesses.

As introduced in Section 4.1, we again use a set of augmentations $\Phi$ to calculate the results of a product of expert ensemble $\overline{P}$. A candidate $s \in C_{p,T}$ is selected for one of the two guesses if it has the (second-)highest probability according to $\overline{P}$. In Figure 5, we compare the rank of the correct solution before and after using this augmentation procedure. In most cases where the correct solution does not start at rank 1, this augmentation leads to a better rank for the correct solution, increasing our chance to solve a given task. In Theorem 4.1, we proved that our product of experts approach is superior to selecting one augmentation at random, which can clearly be seen in Table 3. Here, we compare different sampling methods and different aggregation methods. In *all* cases, using the product of probabilities leads to an increase in score, with $\min P_i$ taking second place for most sampling methods and $\max P_i$ performing the worst. For our $T = 9\%$ DFS inference, PoE increases the final score by $5\%$ compared to averaging the probabilites (66.6% vs 71.6%).

### 5.5. ConceptARC

To make sure we do not overfit on the original ARC data, we further evaluate our method on ConceptARC (Moskvichev et al., 2023) - an ARC-like dataset containing tasks sorted into specific conceptual categories. Our method achieves

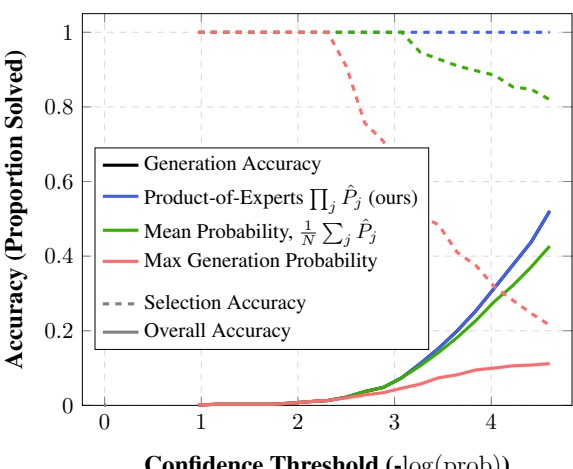

**Accuracy for Different Selection Methods for the 3M Sudoku Dataset (1000 samples)**

*Figure 6.* Results of the Sudoku experiments (plot equivalent to Figure 4, but showing top-1 accuracy instead of top-2). We can see that our product of experts approach increases the accuracy substantially to 53% solved Sudoku puzzles over simply selecting the generated solution with the highest sampling probability. Note that generation accuracy completely coincides with product of experts probability, showing that if a correct solution is sampled, our approach consistently selects it.

73.3% 2-guess accuracy on ConceptARC (using the exact same hyperparameters as DFS T=9%), showing that we generalize well to other ARC-like datasets of similar difficulty.

### 5.6. Sudoku

We further test our approach on the Sudoku 3M dataset (Radcliffe, 2020) to evaluate generalizability of the method to different domains. Since the underlying "rules" of Sudoku remain consistent between tasks, we do not use any test-time training in this case. Instead, we start out with our Llama 3B model pre-trained on ARC, which we then finetune again on 128000 Sudoku tasks. As the Sudoku tasks never have any ambiguity, we report top-1 accuracy rather than top-2. To handle the increased complexity for the LLM compared to ARC, we use DFS with a threshold of T=1%, which provides a good trade-off between accuracy and runtime (see Figure 6). This setup reaches 53% accuracy on 1000 randomly chosen unseen Sudoku puzzles, far better than state-of-the-art LLMs, which have a solve-rate less than 3% on comparable benchmarks (Seely et al., 2025). Notably, *if* the correct solution of a puzzle is sampled, we select it in 100% of cases. This is caused by the fact that Sudoku correctness is simple to evaluate. Using our standard augmentations described in Section 5.3 on the predictions, the model can identify errors more frequently, thereby significantly reducing the likelihood of false positives.

## 6. Discussion

Our method builds on familiar techniques – data augmentation, Bayesian modeling, and product of experts scoring – but tailors them specifically for ARC-like puzzles.

At our methods' core, we use a *single* fine-tuned LLM in two roles: as a *generator*, it proposes solutions for each puzzle augmentation; as a *scorer*, it re-scores each generated candidate across *all* augmentations by taking the product (geometric mean) of likelihoods. The benefit is twofold. First, a candidate solution must be jointly plausible under every valid transformation to rank preferably, making it harder for the model to latch onto spurious correlations found in just one representation. Second, this log-linear pooling approach naturally acts as an ensemble method, as we show in Section 4.1.

Despite ARC's reputation for complexity, our two-phase "generate-then-re-score" routine achieves SOTA results among open models. While only a single closed-source solution (arcprize.org, 2025) posts a higher absolute score at $17 per task, our fully open-source process stands out for its transparency, reproducibility and, above all, its cost-effectiveness of only 0.02$ per task.

By applying these ideas to ARC, we underline a broader principle: when dealing with structured or abstract reasoning tasks, the key factor is to *exploit valid semantic-preserving transformations*, forcing a model to remain consistent across multiple views of the same problem. This allows us to use a single model as an ensemble of experts. We believe this perspective can generalize to more complex symbolic reasoning challenges, wherever such transformations can be defined. Our results demonstrate that large language models, properly steered in inference and supported by prior aware scoring, can go beyond default sampling approaches to capture deeper structures in abstract domains.

### 6.1. Future Work

Building upon our insights, several promising directions emerge for future investigation. First, it would be valuable to further explore the generalizability of using a single large language model as a Product-of-Experts through augmentations beyond ARC-specific transformations. In particular, text-based augmentations such as linguistic reformulations or stylistic variations present possible paths to extend our method to a broader array of natural language reasoning tasks. Second, the effectiveness of our depth-first search (DFS) candidate-generation strategy warrants evaluation beyond ARC-like puzzles; exploring tasks such as logical reasoning, program synthesis, or mathematical problem-solving could yield insights into its broader applicability and effectiveness in structured problem-solving domains.

## Acknowledgement

We would like to express our sincere gratitude to Lambda, for providing computational resources essential for optimizing our pipeline. Specifically, they supplied us with a server equipped with 8xH100 GPUs, enabling rapid iteration on our ideas. Their support was instrumental in winning the ARC Kaggle Competition 2024 using the approach shown in this paper.

This work has been supported by the "Research Center for Algorithmic Intelligence as an Emergent Phenomenon" (funded by the Carl-Zeiss-Stiftung) and by the Deutsche Forschungsgemeinschaft (DFG, German Research Foundation), project 233630050 (Collaborative Research Center TRR 146).

Generative AI language tools were used for text editing; all AI-generated output was subsequently reviewed, revised, and validated by the authors, who assume full responsibility for the final revision.

## Impact Statement

This paper presents work whose goal is to advance the field of Machine Learning. There are many potential societal consequences of our work, none which we feel must be specifically highlighted here.

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

# A. Appendix: DFS Algorithm

The algorithm presented here assumes that the model supports internal caching for already seen sequences and only needs to process the newly added tokens. Note that we use **negative log probabilities** to avoid numerical issues, while the main paper uses percentage values for clarity.

Our actual implementation differs from this simple variant, as we are using *unsloth*, which does not support dynamic caching and requires us to prune the *key-value-cache* of the transformer ourselves.

Furthermore, we use various performance optimizations, like a simultaneous initial forward pass of the best known sequence including prompt and prediction (which is much faster than token-by-token generation) as well as aggregating the sequences during backtracking to avoid the unnecessary processing of sequences that would be discarded later

---

**Algorithm 1** Depth-First Probability-Guided Sampling for LLMs.

---

  **function** DFS_sample $\{model, prompt, threshold, max\_len, eos\_id\}$
    **Input:** $model$ is the language model
    **Input:** $prompt$ is the prompt that should be completed
    **Input:** $threshold$ is the maximum negative log probability allowed
    **Input:** $max\_len$ is the maximum length (including the prompt)
    **Input:** $eos\_id$ is the index of the end of sentence token
    **function** Explore $\{tokens, score\}$
      **if** $tokens[-1] = eos\_id$ **or** $|tokens| \geq max\_len$ **then**
        **return** $(score, tokens)$
      **end if**
      $next\_token\_logits \leftarrow model.predict\_logits(tokens)[-1]$
      $next\_token\_log\_prob \leftarrow -\log\_\text{softmax}(next\_token\_logits)$
      $valid\_sequences \leftarrow \emptyset$
      **for** each possible next token $t$ **do**
        $next\_score \leftarrow score + next\_token\_log\_prob[t]$
        **if** $next\_score \leq threshold$ **then**
          $next\_tokens \leftarrow current\_tokens + [t]$
          $continuations \leftarrow$ Explore$(next\_tokens, next\_score)$
          $valid\_sequences \leftarrow valid\_sequences \cup continuations$
        **end if**
      **end for**
      **return** $valid\_sequences$
    **end function**
    **return** Explore$(prompt, 0.0)$
  **end function**

---

# B. Appendix: Training Parameters

*Table 4.* Training parameters and times for the initial and the test-time fine-tuning processes. Test-time fine-tuning is performed separately for each task, each time starting from the initially fine-tuned base model.

| | Initial Fine-Tuning | Test-Time Fine-Tuning |
|---|---|---|
| Batch size | 4 | 1 |
| Gradient acc. steps | 2 | 1 |
| *LoRA* rank | 256 | 32 |
| *LoRA* $\alpha$ | 24 | 16 |
| *LoRA* bias | off | off |
| rank-stabilized *LoRA* | on | on |
| LR (LoRA adapters) | $1e-4$ | $1e-4$ |
| LR (embeddings) | $1e-5$ | $1e-5$ |
| LR schedule | cosine | cosine |
| LR warmup phase | 25% | 50% |
| Weight decay | off | off |
| Optimizer | *adamw_8bit* | *adamw_8bit* |
| Base model quantization | 4 bit | 4 bit |
| Data type | bfloat16 | bfloat16 |
| Trained tokens | outputs only | outputs only |
| Training dataset | RE-ARC | single task examples |
| Number of Epochs | 368 [Llama] 1200 [NeMo] | 64 |
| Training performed on | 1x Nvidia H100 | 1x Nvidia RTX 4090 |
| Training time | 15 hrs. [Llama] 98 hrs. [NeMo] | 12 sec./task [Llama] 51 sec./task [NeMo] |

*Table 5.* Reduced Token Set for ARC-AGI-specific LLM Model

| Token Category | Tokens | Purpose |
|---|---|---|
| Alphabet | A-Z, a-z (excl. I,O,i,o) | Learned pre-prompt tokens |
| Numbers | 0-9 | Encoding the 10 colors |
| Newline token | \n | Signals end of each grid line |
| Input/Output | I, O | Signals start of problem input/output |
| Begin token | $\langle bos \rangle$ | Inserted once at the beginning |
| End token | $\langle eos \rangle$ | Inserted after each output |
| Padding token | $\langle pad \rangle$ | Internal usage (e.g. batching) |

## C. Appendix: Product of Experts Proof

**Theorem C.1** (Error Bound for Log-Pooled Augmentations)**.** *Suppose we have $m$ valid augmentations $\{\phi_1, \ldots, \phi_m\}$ in the sense of preserving solution distribution, and define*

$$\hat{P}_j(s) := \hat{P}\big(\phi_j(s) \mid \phi_j(p)\big), \text{ for each } j = 1, \ldots, m.$$

*Assume each single-augmentation predictor $\hat{P}_j$ has a bounded KL divergence from P, i.e.,*

$$D_j := \mathrm{KL}\big(P \,\|\, \hat{P}_j\big) \leq \delta_j.$$

*Now define the "geometric-mean" ensemble*

$$\overline{P}(s) := \frac{1}{Z} \prod_{j=1}^{m} \big[\hat{P}_j(s)\big]^{\frac{1}{m}},$$

*where*

$$Z = \sum_{u \in \mathcal{S}_p} \prod_{j=1}^{m} \big[\hat{P}_j(s)\big]^{\frac{1}{m}},$$

*Then the KL divergence between P and $\overline{P}$ is bounded by the* average *of the single-augmentation divergences:*

$$\mathrm{KL}\big(P \,\big\|\, \overline{P}\big) \leq \frac{1}{m} \sum_{j=1}^{m} \mathrm{KL}\big(P \,\|\, \hat{P}_j\big)$$

*Proof.* Let us write

$$D_j = \mathrm{KL}\big(P \,\|\, \hat{P}_j\big) = \mathbb{E}_{s \sim P}\big[-\log \hat{P}_j(s)\big] - \mathbb{E}_{s \sim P}\big[-\log P(s)\big].$$

By assumption, $D_j \leq \delta_j$ for each $j$.

**Step 1: Expressing $\mathrm{KL}\big(P \,\|\, \overline{P}\big)$.** By definition of KL divergence,

$$\mathrm{KL}\big(P \,\|\, \overline{P}\big) = \sum_{s \in \mathcal{S}_p} P(s) \log\big(\tfrac{P(s)}{\overline{P}(s)}\big)$$

$$= \mathbb{E}_{s \sim P}\big[-\log \overline{P}(s)\big] - \mathbb{E}_{s \sim P}\big[-\log P(s)\big].$$

Since we can rewrite $\overline{P}$ as

$$\overline{P}(s) = \frac{1}{Z} \exp\Big(\tfrac{1}{m} \sum_{j=1}^{m} \log \hat{P}_j(s)\Big),$$

we get

$$-\log \overline{P}(s) = -\tfrac{1}{m} \sum_{j=1}^{m} \log \hat{P}_j(s) + \log Z.$$

Thus,

$$\mathbb{E}_{s \sim P}\big[-\log \overline{P}(s)\big] = \frac{1}{m} \sum_{j=1}^{m} \mathbb{E}_{s \sim P}\big[-\log \hat{P}_j(s)\big] + \log Z.$$

Subtracting $\mathbb{E}_{s \sim P}[-\log P(s)]$ then yields

$$\mathrm{KL}\big(P \,\|\, \overline{P}\big)$$

$$= \underbrace{\frac{1}{m} \sum_{j=1}^{m} \Big[\mathbb{E}_{s \sim P}\big(-\log \hat{P}_j(s)\big) - \mathbb{E}_{s \sim P}\big(-\log P(s)\big)\Big]}_{\frac{1}{m} \sum_{j=1}^{m} \mathrm{KL}(P \,\|\, \hat{P}_j)}$$

$$+ \log Z.$$

Hence to complete the bound, we need only to show that $\log Z \leq 0$, i.e. that $Z \leq 1$.

**Step 2: Bounding** $\log Z$**.** Recall that

$$Z = \sum_{s \in \mathcal{S}_p} \prod_{j=1}^{m} [\hat{P}_j(s)]^{\frac{1}{m}}.$$

Since the geometric mean is always smaller than the arithmetic mean for positive numbers, it follows that:

$$Z \leq \sum_{s \in \mathcal{S}_p} \sum_{j=1}^{m} \frac{1}{m} [\hat{P}_j(s)]$$

with equality exactly when all $\hat{P}_j$ are equal. Further, as all $\hat{P}_j$ are probability distributions we find that:

$$Z = \sum_{s \in \mathcal{S}_p} \sum_{j=1}^{m} \frac{1}{m} [\hat{P}_j(s)] = \frac{1}{m} \sum_{j=1}^{m} \sum_{s \in \mathcal{S}_p} [\hat{P}_j(s)] \leq 1$$

**Putting it all together.** From Step 1 of the proof, we have the decomposition

$$\mathrm{KL}(P \,\|\, \overline{P}) = \underbrace{\frac{1}{m} \sum_{j=1}^{m} \mathrm{KL}(P \,\|\, \hat{P}_j)}_{\text{average excess NLL}} + \log Z.$$

Combining with the bound $Z \leq 1$ yields

$$\mathrm{KL}(P \,\|\, \overline{P}) \leq \frac{1}{m} \sum_{j=1}^{m} \mathrm{KL}(P \,\|\, \hat{P}_j)$$

□

