# OpenReview forum: "Product of Experts with LLMs: Boosting Performance on ARC Is a Matter of Perspective"
_ICML.cc/2025/Conference — ICML 2025 poster_

### Official Review · Reviewer_2b2p · 2025-03-13

**Overall Recommendation:** 4

**Summary:**

This paper proposes a new way to solve the ARC tasks. It first employs a depth-first search algorithm to generate diverse, high-probability candidate solutions for the ARC tasks, then applies an LLM to not only act as a generator but also as a scorer, using its output probabilities to select the most promising solutions. Experimental results show that their proposed method can enhance LLM's performance on ARC.

## update after rebuttal

In the rebuttal, the authors have addressed my previous concerns and thus i have updated my score correspondingly.

**Claims And Evidence:**

Please see the weakness section.

**Essential References Not Discussed:**

N/A

**Experimental Designs Or Analyses:**

Yes, the experiments are generally sound.

**Methods And Evaluation Criteria:**

Yes, the proposed methods make sense for the problem.

**Other Comments Or Suggestions:**

Please see the weaknesses above.

**Other Strengths And Weaknesses:**

Weaknesses:
1. Line 252-254: why autoregressive will lead to this problem? please explain it.
2. in section 5.1, the authors mention that they limit the number of tokens in the vocabulary to 64, which may limit the generalizability of the fine-tuned models from ARC to other tasks.

**Questions For Authors:**

1. Line 252-254: why autoregressive will lead to this problem? please explain it.
2. I understand that ARC is an important (inductive) reasoning task. However, have the authors tried their methods on other inductive reasoning tasks? I believe that this will make the proposed method more useful for the whole community.

**Relation To Broader Scientific Literature:**

The key contributions of the paper are related to the broader scientific literature.

**Theoretical Claims:**

Yes, I have checked the claims.

---

> ### Author Rebuttal · Authors · 2025-03-31
>
> We thank the reviewer for their time and review, and we address each of the points raised below and improve the clarity of the manuscript in the next revision.
>
> Regarding why the autoregressive nature leads to the mentioned problem: (from the paper: “However [...] the highest probability solution according to a single augmentation is not always correct, partly due to autoregressive inconsistencies.”)
> The claim in question was that in autoregressive models, the highest probability sentence is not necessarily the correct one. This becomes qualitatively more evident in the additional Sudoku experiment that we have expanded for this discussion (see below), which might help explain this more clearly: The autoregressive LLM can only attend to tokens in the past to reason what the next one should be. Because of this, it is sometimes necessary to make decisions that depend on information that becomes available later, even if the problem might be simple. For example, consider a Sudoku where the first cell is empty. In some cases the LLM would need to solve the full Sudoku internally before making the first prediction. As a result, errors can be introduced with relatively high confidence. After this error has been predicted, at some point there is no possible correct prediction that can be made. At this point the LLM behaviour is effectively undefined, as it is not trained to be stable under prediction errors. In other words, follow-up errors do not necessarily have a low probability, and the first error can be caused by sheer complexity of the necessary prediction due to the autoregressive nature. We will aim to explain this more clearly in the next revision.
>
> Regarding limiting our tokens:
> Limiting the tokens indeed limits the trained model's generalizability to other tasks, but the methodological approach is not tied to the small vocabulary. Instead we reduced the vocabulary for pragmatic reasons (to reduce the memory consumption in the spirit of the ARC Challenge) as the input and output embedding layers contain a very large number of weights and tokenization merges can substantially inhibit reasoning on numbers. Of course, instead of replacing the tokenizer, additional tokens could be used instead.
>
> Regarding generalization:
> We agree that extending our approach and demonstrating its generalizability beyond ARC-like problems is important, and this is an active area of work for us. While many reasoning datasets pose challenges due to reliance on language output (where multiple correct answers can dilute probability), Sudoku offers a strong test case for structured reasoning. To provide evidence for broader applicability, we enhanced our preliminary Sudoku experiments significantly.
>
> **Extended Sudoku Experiments**:
>
> We fine-tuned a Llama 3B model on 1 million puzzles from the 3M Sudoku dataset and applied our method to solve hard instances (requiring 19 to 31 clues). Using 8 augmentations and a 1% DFS threshold (with consistent hyperparameters otherwise), our pipeline correctly solved 52% of 1000 test Sudokus using a single guess. (Figure 4 adapted for Sudoku, illustrating this experiment, is available at https://imgur.com/a/SY4g2OQ). This is a notable success on a task known to be very difficult for LLMs. Importantly, the Product-of-Experts component was highly reliable, identifying the correct solution in 100% of the cases, where the correct solution was among the candidates generated by DFS (blue dotted line). We believe this result is complementary to our main results and shows that augmentation scoring using a single LLM can substantially increase the performance of an LLM on reasoning tasks.

---

### Official Review · Reviewer_SUoh · 2025-03-14

**Overall Recommendation:** 4

**Summary:**

This paper presents a novel approach to solving the Abstraction and Reasoning Corpus (ARC-AGI) challenge, which tests abstract reasoning abilities in AI systems. The authors achieve SOTA performance for open-source models with a 71.6% accuracy rate (286.5/400 solved tasks) on the public ARC-AGI evaluation set.
Key Contributions:

- DFS-based sampling approach to generate diverse, high-probability candidate solutions.
- Product of Experts (PoE) Scoring: The model serves as both a generator and scorer, using its output probabilities across different augmentations to select the most promising solutions.
- Data augmentation approach: task-specific data augmentations throughout training, generation, and scoring phases, including rotations, reflections, color permutations, and example reordering.

## Update after rebuttal

The authors answered my questions with great detail and provided convincing elements to address my concerns. I have therefore updated my rating to accept.

**Claims And Evidence:**

Nothing to report.

**Essential References Not Discussed:**

Nothing to report.

**Experimental Designs Or Analyses:**

Nothing to report.

**Methods And Evaluation Criteria:**

ARC-AGI is the program synthesis benchmark reference to measure reasoning abilities.

**Other Comments Or Suggestions:**

None

**Other Strengths And Weaknesses:**

- Why BFS vs DFS? Ablation on that?
- Any sign of overfitting to re-arc? How much data was used? Were new tasks randomly generated at each epoch or was a fixed dataset used?

**Questions For Authors:**

1. Why DFS and not BFS? Are there any ablations on that?
2. Is there any sign of overfitting to re-arc? How much re-arc data was used? Were new tasks randomly generated at each epoch or was training done on a fixed dataset sampled from re-arc once and for all the training?

**Relation To Broader Scientific Literature:**

Nothing to report.

**Theoretical Claims:**

Nothing to report.

---

> ### Author Rebuttal · Authors · 2025-03-31
>
> We thank the reviewer for their positive review.
>
>
> **Regarding DFS vs BFS**:
>
> The output of our generation process would be the same if we use DFS or BFS, as only the paths with sufficiently high sampling probability are kept. However, using BFS would make our optimizations harder. During generation, transformers use a key-value cache of previously generated tokens to avoid recalculation of this data. DFS has the advantage that we need to keep this cache for only one path in memory, thereby requiring only about as much memory as standard sampling. For BFS, on the other hand, we would need to keep this cache in memory concurrently for all parallel paths that are being explored, which would require a lot of memory (though on very large systems, one might be able to parallelize the inference better when using BFS). In our experiments, we chose to evaluate Beam Search instead, which is essentially a width-constrained version of BFS, and follows only the most promising paths to reduce the memory footprint.
>
>
> **On ReARC**:
>
> We have seen no signs of overfitting on ReARC, and our ReARC trained model also performs well on other datasets besides the ARC eval set. For our largest model, we trained for 1200 epochs, using 6 examples (+1 challenge) in each prompt. As we never re-used any of the generated re-arc examples, this required 8400 examples for each or the 400 tasks, which we generated with the re-arc github code. We also evaluated the model using only test-time training on the ConceptARC dataset and achieved similar results to the ARC evaluation set (reaching 73.3% top 2 accuracy with DFS-9%), indicating that there is no overfitting specifically to the official ARC datasets. Additionally, we ran our method on Sudoku tasks, which have so far proven very hard for LLMs to solve. Our method allows a fine-tuned LLM to solve 52% of 1000 random Sudokus drawn from the 3M Sudoku dataset. We uploaded Fig. 4 adapted for Sudoku to https://imgur.com/a/SY4g2OQ.

---

### Official Review · Reviewer_LUdW · 2025-03-14

**Overall Recommendation:** 4

**Summary:**

The paper describes a system for the ARC challenge, based on using data augmentations.
The augmentations are basic transformations of the images (rotation, reflection, shuffling of the example order and permutation of colors) that can be applied to both the input problem and the solution, such that the transformed solution corresponds to the transformed problem.
These augmentations are then used for:
1) Generating additional data for regular supervised training of the model.
2) Generating alternative solution candidates. A given problem is transformed with different augmentations to produce alternative outputs, which can then be un-transformed to get alternative solutions to the original problem.
3) Reranking of the candidate solutions. The probability of each candidate solution is calculated under each possible transformation, then combined to get an overall score for that solution.

Test-time training from previous work is also applied.
The results show top open-source results (71.6%) on the public ARC evaluation set, with only GPT 03 outperforming it.

**Claims And Evidence:**

The work is interesting.
The ideas are good, showing creative ways of how task-specific augmentations can be used to improve performance and search the candidate space.
The results are good, setting a new state-of-the-art for open-source ARC models.

**Essential References Not Discussed:**

.

**Experimental Designs Or Analyses:**

Beam search is shown to perform just as well as DFS, while the advantage of DFS is speed.
These approaches are fundamentally doing the same thing - generating alternative candidates while pruning possible paths.
Please provide more of an explanation about where the speed advantage of DFS is originating from (or performance advantage, when considering comparable speed).

**Methods And Evaluation Criteria:**

Yes

**Other Comments Or Suggestions:**

Typo:
"make us of alternative" -> "make use of alternative"

**Other Strengths And Weaknesses:**

The clarity of the paper and some details could be improved.
Currently, different graphs and tables are provided but only mentioned very briefly much later in the paper, without much analysis.
The problem is formulated in a bayesian framework but it is unclear why that is necessary or what benefit it provides.
There are various other open questions that need clarification (below).

It is difficult to understand what Figure 4 is showing exactly. It could use some more explanation.

On page 4 there is a large section of sampling under the "Training" sub-section. Not sure why that is there, it should probably belong to the "Candidate Generation" subsection.

Are the augmentations used during test-time training as well?

"As illustrated in Figure 2, we add a small number of extra tokens to the start of a task."
It is not clear what these tokens are. Do you mean the "A...Za...z" tokens in Figure 2? If so, it is not clear what the actual contents (text?) of those tokens is.

DFS generates candidates that can then be ranked using PoE. In Table 2, it is unclear how you rank candidates with PoE without the candidate generation step (DFS).

When using beam search instead of DFS, it is unclear whether beam search also makes use of augmentations or is it just searching alternative outputs for a single input.

Beam search is shown to perform just as well as DFS, while the advantage of DFS is speed.
These approaches are fundamentally doing the same thing - generating alternative candidates while pruning possible paths.
Please provide more of an explanation about where the speed advantage of DFS is originating from (or performance advantage, when considering comparable speed).

Algorithm 1 in the appendix seems confusing:

1. The DFS threshold in the main paper is defined as a percentage (e.g. 9%). It is not clarified what this represents or how it is applied.
In Algorithm 1 the threshold seems to be an absolute logprob threshold instead.

2. The algorithm continues as long as the current score is smaller than the threshold. While adding logprob values, the score can only get smaller. Should this be greater than the threshold instead?

3. Assuming that this is greater-than the threshold instead, the score still doesn't seem to be normalised by length anywhere and it will continuously get smaller as more tokens are generated. Is that not an issue?


####### update after author response
Thank you for your responses. I am already quite positive with the score and I will stick to it.

**Questions For Authors:**

.

**Relation To Broader Scientific Literature:**

SOTA compared to other open-source solutions to this problem

**Theoretical Claims:**

.

---

> ### Author Rebuttal · Authors · 2025-03-31
>
> We thank the Reviewer for their careful evaluation, interesting questions and positive feedback. We appreciate the detailed questions and suggestions, which will help us improve both the clarity and presentation of our work. Below, we address each of the reviewer’s points:
>
> Bayesian Framework: We employed a Bayesian formulation to rigorously detach the uncertainty of the problem from the uncertainty of the erroneous LLM prediction for a single augmentation. Although this formulation did not alter our pipeline, it provides a formal grounding for ARC-like challenges and helps understand the theoretical value that the PoE approach has and shows that it is guaranteed to perform well under reasonable assumptions.
>
> We agree that Figure 4 needs a more detailed explanation. We will revise the caption and discussion to clearly describe what is being illustrated. To clear any potential misunderstandings:
> Solid colored lines denote the number of tasks solved using a specific selection algorithm. The solid black line shows the number of tasks where the correct solution was amongst the sampled candidates, and thereby provides an upper bound for the performance of the selection algorithms. The dotted lines evaluate the performance of our selection algorithms, compared to this upper bound: What percentage of correct candidates are actually selected when they are present? It shows that even when using low DFS probabilities -  and therefore sampling a high number of candidates - PoE is able to select the correct solution among all candidates with high specificity.
>
> Regarding augmentations during Test-Time training and beam search comparisons: The same type of augmentations is consistently applied, however, we switch color and example permutations randomly whenever a task is drawn. We will clarify this in the revised version to avoid any ambiguity.
>
> To clarify the use of augmentations in the ablation study without DFS: In this case we simply use stochastic sampling for each of the 16 different augmentations, thus generating up to 16 different candidates, which we will also clarify in the revised version.
>
> The extra tokens mentioned (depicted as “A-Za-z” in Figure 2) are tokens representing single letters (‘A’, ‘B’, ‘C’...). As we also train the embeddings during fine-tuning, they take the role of a soft-prompt and are trained along with the model parameters. We will provide a more detailed explanation of their content and function and provide an additional ablation study regarding their usage in the next revision.
>
> Regarding the dfs threshold as used in the algorithm: We will change this to be more clear. We decided to use percentages in the main paper as they are easier to understand, but algorithmically we add (negative) logprobs instead to mitigate floating-point errors. This also answers the second point regarding threshold comparison.
>
> On normalization by length, we have considered it, but surprisingly, this was rarely an issue for the ARC tasks. Investigating this, we found that the LLM has very high certainty for almost all positions, only being uncertain at some key pixels. Further, we found that the generated candidates rarely differ in length, as the model is very good at predicting the correct output shape, mitigating the effect of length normalization.
>
> We will incorporate the noted typesetting-related suggestions in the next revision.
>
> **Reasons why DFS is faster than beam search**:
>
> The speed advantage of the DFS algorithm during the candidate generation step comes mostly from the early pruning of low-probability paths. In Beam Search, the same amount of paths is explored each time, regardless of their cumulative sampling probability. As DFS stops when the cumulative sampling probability falls beyond the threshold, low-probability paths will not be explored by DFS, thereby reducing unnecessary computations. For example, in cases when the correct solution has a sampling probability higher than (1 - threshold), which happens frequently for the simpler tasks, DFS is guaranteed to follow only a single path and explores no side-branches at all, making it as fast as standard sampling of a single candidate.
>
> Additionally, for all subsequent augmentations after the first one, we pass the most promising candidate solution as an initial guess to the DFS, and change the DFS exploration order to process this sequence first in a single forward pass of the model, which is much faster than token-by-token generation. Note that this does not change the result of the DFS algorithm, as all paths are still being explored.
>
> As the DFS prunes low-probability solutions, the average number of candidates generated per task is also much lower for DFS with T=9% than with 4x Beam search, thus also making the subsequent scoring process faster as fewer candidates have to be scored. Note that the advantage during scoring could be mitigated by post-filtering the beam search results with a similar probability threshold.

---

### Official Review · Reviewer_Bm9D · 2025-03-14

**Overall Recommendation:** 3

**Summary:**

Authors propose a new approach to solve ARC-AGI challenge. In particular, authors train an LLMs to generate diverse, high probability solutions using augmented training data. Authors define an augmentation transformation for ARC-AGI dataset, which include rotations
and reflections of each task, shuffling of the example order and permutation of colors. Next, authors systematically explore the space of solutions via a depth first search (DFS) algorithm, pruning any partial path whose accumulated probability falls below dynamically updated threshold. Finally, the solution is selected based on a product of probabilities across all augmentations. Authors test their approach of two models, and ablate on each component of their algorithm, showing the effectiveness of the proposed approach.

## update after rebuttal
I appreciate detailed response provided by authors. The paper has merits, but my main concern rating generalizability is still open. I will keep my score as weak accept

**Claims And Evidence:**

Yes

**Essential References Not Discussed:**

N/A

**Experimental Designs Or Analyses:**

Yes, experiments described in Sec 5.
Generalizability of the proposed approach remains unexplored. Although authors provide preliminary results in Appendix A1 for Sudoku task, more systematic experiments are needed.

**Methods And Evaluation Criteria:**

Yes, authors focus on a single benchmark, and test their approach with two opensource models.

**Other Comments Or Suggestions:**

Lines 085-085: "The objective of the individual ARC tasks is to discern this rule" - which "this rule"?
Line 192: "The trained language model (LLM)" - LLM already defined in line 203
Line 226: "This two-step approach—(1) DFS-base" - spaces missing around "-"?

Overall, formatting seems off: some paragraphs have space between each other - which is expected - but some not. For example, lines 283-300, right column.  Line numbering on page 5 is suddenly cursive. Appendices are not named as such in sections after references, just some tables and pictures are dumped together. Should be fixable by following the template proposed by organizers: https://icml.cc/Conferences/2025/CallForPapers

**Other Strengths And Weaknesses:**

Paper is clear and well-written, although with some formatting issues. Proposed approach is novel and intricate. However, generalizability of the proposed approach remains unexplored. Although authors provide preliminary results in Appendix A1 for Sudoku task, more systematic experiments are needed.

**Questions For Authors:**

See "Other Strengths And Weaknesses"

**Relation To Broader Scientific Literature:**

Authors focus on ARC-AGI benchmark, that seeks to measure generalization on novel tasks, as opposed to other popular benchmarks that skill at tasks that can be prepared for in advance. Authors elaborate on test-time-training approach, utilizing task-specific augmentation transformation to synthesize more training data, and evaluate candidate solutions.

**Theoretical Claims:**

No

---

> ### Author Rebuttal · Authors · 2025-03-31
>
> We thank the reviewer for their time and valuable feedback. We appreciate the detailed comments, and we address each of the points raised below.
>
> First of all, we appreciate the reviewer’s careful observations regarding formatting issues. We will revise the manuscript to address inconsistencies such as inconsistent paragraph spacing and other typographic details by following the organizers’ template.
>
> We would like to clarify that the threshold used for candidate generation was not dynamically updated during candidate selection. Instead, it was chosen as a trade-off between computational speed and solution quality. We agree that a dynamically tuned threshold could potentially improve performance and view this as an interesting direction for future work.
>
> We agree that assessing the generalizability of our approach is crucial. The focus on the ARC-AGI benchmark was intentional, as it emphasizes generalization on novel tasks using minimal example pairs. Nonetheless, a more systematic evaluation, namely extending to other tasks and benchmarks, is very important. In our revised manuscript, we will elaborate on potential future work, including: (1) discussing the idea of using a single LLM as a Product-of-Experts via augmentations on language tasks, which could also be explored using text-based augmentations such as reformulations or style changes. (2) testing the DFS component in generating high-quality candidates for tasks beyond ARC-like problems.
>
> In response to concerns regarding the systematic evaluation of generalization, we have prepared a more detailed analysis of the Sudoku experiment (we uploaded Fig. 4 adapted for Sudoku to https://imgur.com/a/SY4g2OQ). This task, which shares characteristics with ARC in terms of example pairs, 2D data, and logical reasoning, provides further insights into the potential and limitations of our approach. Here, we achieve 53% solve rate for Sudokus using a fine-tuned Llama 3B model. (Please refer to our answer to Reviewer 2b2p for more details.) Finally, we also apply our method to the ConceptARC dataset, achieving similar results (73.3% accuracy with DFS-9% and otherwise same hyperparameters). While this dataset has a lot of similarity to the ARC dataset, it still provides some indication of generalization.

---

### Decision · Program_Chairs · 2025-05-01

**Decision:**

Accept (poster)

**Comment:**

The authors propose to sovle the ARC-AGI challenge through training an LLM to generate diverse set of candidate solutions searched over through DSF. Almost sota scores are reached. Reviewers agree on the paper being well written, significant novelty and good empirical results. The rebuttals are acknowledged. We urge the authors to take into account the minor writing criticism and improve their paper. Other than that the paper is a good contribution and most suitable for publication at ICML.